# Drive Anywhere: Generalizable End-to-end Autonomous Driving with Multi-modal Foundation Models

**Tsun-Hsuan Wang**[1]**, Alaa Maalouf**[1]**, Wei Xiao**[1]**, Yutong Ban**[2]**,**
**Alexander Amini**[1]**, Guy Rosman**[3]**, Sertac Karaman**[4]**, Daniela Rus**[1]
[1]MIT CSAIL, [2]Shanghai Jiao Tong University, [3]TRI, [4]MIT LIDS
https://drive-anywhere.github.io

**Abstract:** As autonomous driving technology matures, end-to-end methodologies have emerged as a leading strategy, promising seamless integration from perception to control via deep learning. However, existing systems grapple with challenges such as unexpected open set environments and the complexity of black-box models. At the same time, the evolution of deep learning introduces larger, multimodal foundational models, offering multi-modal visual and textual understanding. In this paper, we harness these multimodal foundation models to enhance the robustness and adaptability of autonomous driving systems. We introduce a method to extract nuanced spatial features from transformers and the incorporation of latent space simulation for improved training and policy debugging. We use pixel/patch-aligned feature descriptors to expand foundational model capabilities to create an end-to-end multimodal driving model, demonstrating unparalleled results in diverse tests. Our solution combines language with visual perception and achieves significantly greater robustness on out-of-distribution situations. Check our website (https://drive-anywhere.github.io) for more videos and demos.

**Keywords:** End-to-end Driving, Generalization, Foundation Models

## 1 Introduction

In this work, we aim to harness the power of multimodal foundation models to enhance the generalization and reliability of end-to-end autonomous driving systems. Importantly, rather than relying on explicitly-defined data formats like scene descriptions or segmentation maps, we exploit the latent features at model inference to preserve all information pertinent to the model's reasoning process.

**Foundation models as feature extractors**. While these models exhibit certain favorable characteristics for attaining open-set, multimodal representations, they do not translate seamlessly to autonomous driving. The significant constraint arises from the fact that these models are primarily designed for image input consumption, resulting in the generation of a singular vector representation for the entire image within an embedding space. However, decision-making in autonomous driving demands more than just semantic scene descriptions; it also requires nuanced spatial and geometric information. To address this, we present a generic method to extract per-patch features from transformer-based architectures that is broadly applicable to a wide range of foundation models.

**Simulation using language.** Multimodal representations map data from various modalities into a unified embedding space, offering two key advantages for policies trained to operate in this space: (i) cross-modality feature inspection, and (ii) feature manipulation in modalities distinct from sensor measurements. These capabilities facilitate latent space simulation, allowing target features tied to specific concepts to be swapped out with features from other desired concepts. For instance, one could replace a *car* feature from images with a *deer* feature, without requiring sensor data synthesis for a deer. This method is valuable for data augmentation and policy debugging. Notably, leveraging the language modality as a conceptual representation enables integration with large language models (LLMs), unlocking enhanced common-sense reasoning capabilities to enrich simulation complexity.

7th Conference on Robot Learning (CoRL 2023), Atlanta, USA.

**Our contributions.** We bridge the gap between the robust multimodal open-set capabilities demonstrated by foundation models and the advanced reasoning capabilities expected of futuristic autonomous systems - enabling OOD, end-to-end, multimodal, and more explainable autonomy:

- A novel mechanism to extract pixel/patch-aligned features, extending the capabilities of multimodal foundation models that typically yield image-level vectors.

- A latent space simulation technique augmented with language modality for both data augmentation in training and counterfactual reasoning in policy debugging.

- Extensive analysis in photo-realistic simulated environments to demonstrate enhanced generalization across diverse scenarios (scenes not seen during training and obstacles not trained on).

- Deployment and validation on a full-scale autonomous vehicle in real-world environments.

## 2   Method

**Patch-wise Feature Extraction.** Given a foundation model $\texttt{Desc} : \mathbb{R}^{H \times W \times 3} \to \mathbb{R}^D$ of $L$ layers, an input image/frame $F \in \mathbb{R}^{H \times W \times 3}$, and a desired resolution $H' \leq H, W' \leq W$, the goal is to extract a feature descriptors tensor $F' \in \mathbb{R}^{H' \times W' \times D}$, such that $F'$ encapsulates all the semantic information of $F$ and maintains its location in the scene. For simplicity, we set $H', W'$ to be equal to the number of (non-overlapping) patches used to divide the input image $F$ when applying $\texttt{Desc}$ on it (in what follows we will show how $H', W'$ can be any number smaller than $H, W$) and $N = H'W'$. For an integer $i > 1$, we use $[i]$ to denote the set $\{1, \cdots, i\}$, and for every layer $\ell \in [L]$, we use $Q^\ell_{\texttt{Desc}(F)}, K^\ell_{\texttt{Desc}(F)} \in \mathbb{R}^{N \times D_k}, V^\ell_{\texttt{Desc}(F)} \in \mathbb{R}^{N \times D}$ to denote the resulted query, key, and value matrices in the $\ell$th attention layer, when applying $\texttt{Desc}$ on $F$. We provide a mechanism to extract features for a specific patch (or area in the image) $F'^{(j)}$, where $j \in [N]$. Notably, this mechanism can be applied at any layer of most transformer-based models such as CLIP [1], DINO [2, 3], and BLIP [4, 5].

When extracting $F'^{(j)}$, we introduce an attention mask $m^{(j)} = (m^{(j)}_1, \cdots, m^{(j)}_N) \in \mathbb{R}^N$. Each element $m^{(j)}_i \in [0, 1]$ of this vector determines how much the $i$th patch should contribute to the desired patch feature $F'^{(j)}$. For example, if we want to completely ignore patch number $i$, simply set $m_i = 0$ and $m_k = 1, \forall k \in [N] \backslash i$. We utilize $m$ to extract features as follows:

(1) Set $r \in (-\infty, 0)$ as the parameter to control the strength of the masking; the larger $|r|$, the higher effects of the masking.

(2) Define the matrix $G^\ell_{\texttt{Desc}(F)}$ as the matrix multiplication of the key and query matrices at the $\ell$th attention layer:

$$G^\ell_{\texttt{Desc}(F)} := Q^\ell_{\texttt{Desc}(F)} (K^\ell_{\texttt{Desc}(F)})^T.$$

(3) Given the matrix $M^{(j)} = [m^{(j)}, \cdots, m^{(j)}]^T \in \mathbb{R}^{N \times N}$, we obtain a masked version of $G^\ell_{\texttt{Desc}(F)}$ as:

$$\hat{G}^{\ell,(j)}_{\texttt{Desc}(F)} = G^\ell_{\texttt{Desc}(F)} + (\mathbf{1} - M^{(j)}) \cdot r,$$

where $\mathbf{1} \in \mathbb{R}^{N \times N}$ is an all-ones matrix. This operation sets the attention scores (in the matrix $\hat{G}^{\ell,(j)}_{\texttt{Desc}(F)}$) for "non-contributing" patches (where there corresponding $m_i$ close to 0) to be close to $r$ (low value), effectively masking them out. The $\mathbf{1} - M^{(j)}$ term ensures that the patches with a corresponding attention mask equal to 1 have an added softmax'ed score of 0 (no modification), and a very low value (effectively $r$) if the corresponding attention mask is near 0.

(4) With the modified attention scores, we obtain the final attention weights as:

$$F'^{(j)} := \texttt{Desc}^{\ell \to} \left( \text{SoftMax}(\hat{G}^{\ell,(j)}_{\texttt{Desc}(F)}) (V^\ell_{\texttt{Desc}(F)})^T \right),$$

where $\texttt{Desc}^{\ell \to}$ is the rest of the model after the $l$th layer. Notably, this technique can be extended to region-wise feature extraction by generalizing the definition of patches to arbitrarily-shaped regions.

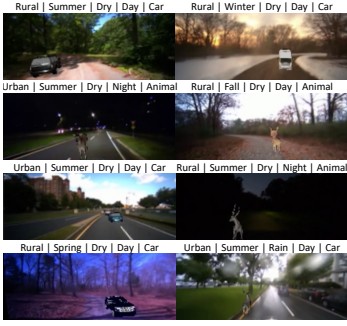

Table 1: **OOD generalization.** The left figures are illustrations for the scenarios. †indicates car types different from training. *ID* is in distribution. *OOD* is out-of-distribution.

| Setting | Scene | Season | Scenarios Weather | Time | Actor | No-FM | Methods MF | I-ViT | Ours |
|---|---|---|---|---|---|---|---|---|---|
| ID | Rural | Summer | Dry | Day | Car | 1.00 | 0.72 | 1.00 | 1.00 |
| OOD | Rural | Spring | Dry | Day | Car† | 0.84 | 0.42 | 0.86 | 0.96 |
| | | Summer | Dry | Night | Car† | 0.30 | 0.35 | 0.80 | 0.89 |
| | | Fall | Dry | Day | Car† | 0.90 | 0.74 | 0.95 | 0.91 |
| | | Winter | Snow | Day | Car† | 0.14 | 0.42 | 0.88 | 0.96 |
| | | Spring | Dry | Day | Animal | 0.85 | 0.39 | 0.89 | 0.95 |
| | | Summer | Dry | Night | Animal | 0.29 | 0.39 | 0.59 | 0.85 |
| | | Fall | Dry | Day | Animal | 0.87 | 0.71 | 0.95 | 0.88 |
| | | Winter | Snow | Day | Animal | 0.15 | 0.45 | 0.87 | 0.95 |
| | Urban | Summer | Dry | Day | Car† | 0.55 | 0.50 | 0.77 | 0.62 |
| | | Summer | Rain | Day | Car† | 0.69 | 0.43 | 0.81 | 0.81 |
| | | Summer | Dry | Night | Car† | 0.45 | 0.42 | 0.81 | 0.78 |
| | | Summer | Dry | Day | Animal | 0.58 | 0.50 | 0.80 | 0.64 |
| | | Summer | Rain | Day | Animal | 0.66 | 0.43 | 0.83 | 0.78 |
| | | Summer | Dry | Night | Animal | 0.45 | 0.36 | 0.86 | 0.81 |

**Language-augmented Latent Space Simulation.** Each patch feature $F^{'(j)}$ incorporates language modality, enabling seamless integration with LLMs. We exploit this property by conducting latent space simulations, where we replace $F^{'(j)}$ with alternative textual features to simulate different scenarios. We opt for feature replacement over arithmetic operations as the latent space may not necessarily adhere to a Euclidean metric structure. The procedure is as follows:

(1) Obtain a set of concepts in natural language that may be relevant to autonomous driving from LLMs and compute their corresponding textual feature,

$$T_k = \mathrm{Desc}(c_k), \text{ where } c_k \in \mathtt{C}^{\mathrm{src/tgt}} = \mathrm{LLM}(\langle questions \rangle)$$

where $\mathtt{C}^{\mathrm{src}}$ is the set that may appear in the image feature and $\mathtt{C}^{\mathrm{tgt}}$ is the set of the desired substitutes.

(2) Find the best match of the patch feature via search with similarity measure $g(\cdot, \cdot)$,

$$T_{F'^{(j)}} = \underset{k \in [|\mathtt{C}^{\mathrm{src}}|]}{\arg\max} \, g(F^{'(j)}, T_k)$$

This step can be improved by more advanced techniques like text inversion [6] or prompt tuning [7].

(3) Manipulate the dense feature descriptor $F'$ by replacing $F^{'(j)}$ with sensible textual features $h(T_{F'^{(j)}}, \{T_k\}_{k \in [|\mathtt{C}^{\mathrm{tgt}}|]})$ under conditions like similarity above certain threshold or stochasticity. The function $h$ can be human prior or LLMs that conceptually answer the question of *what may be a plausible substitute from $\mathtt{C}^{\mathrm{tgt}}$ under the current context*.

## 3  Experimental Results

**Out-of-Distribution Generalization.** In Tab. 1, we evaluate the out-of-distribution (OOD) generalization capabilities of end-to-end policies employing various feature extractors. Our baseline comparisons include: (i) *No Foundation Model (No-FM)* [8, 9], which utilizes a CNN-based model (transformer-based architectures yielded comparable results) trained from scratch without leveraging foundation models; (ii) *Mask-based Features (MF)* [10, 11], which initially applies segmentation [12], then extract global feature vectors [1] for each masked/cropped image, and assign the vector to each pixel within the corresponding region (an approach adapted from similarity measure technique); (iii) *Inherent ViT Features (I-ViT)* [13], which suggests using ViT models[14] interlayers outputs (per-patch corresponding) of the key, value, and query matrices as "inherent" per-pixel/patch features. All feature extractors are followed by a policy network utilizing a consistent transformer-based architecture. Firstly, we note that *MF* underperforms in both in-distribution and out-of-distribution settings. We propose two possible explanations for this: (1) the process of masking out non-target regions may inadvertently eliminate valuable contextual information, and (2) while masking is generally effective for objects, it creates ambiguous image crops for more abstract categories, often referred to as "stuffs" [15]. For instance, masking out all elements except the road in a rural setting results in an indistinct, yellowish region. Additionally, *I-ViT* and *Ours* surpass *No-FM* in OOD settings, scenarios, highlighting the benefits of utilizing foundation models as feature extractors to enhance generalization. Note that *I-ViT* does not incorporate language modality.

| RSDDC | RSDNC | RFDDC | RWSDC | RSDDA |
|-------|-------|-------|-------|-------|
| +3.43% | -2.49% | +8.32% | +3.12% | -0.48% |
| **RSDNA** | **RFDDA** | **RWSDA** | **USDDC** | **USRDC** |
| -5.09% | +9.83% | +1.02% | +12.49% | +14.44% |
| **USDNC** | **USDDA** | **USRDA** | **USDNA** | **All** |
| +10.80% | -0.65% | +13.08% | +12.75% | **+5.47%** |

Table 2: **Improved generalization from data augmentation**. We augment training with unseen yet potentially relevant concepts from LLMs via language-augmented latent space simulation to improve performance. (The labels follow Tab. 1, e.g., *RSDDC* is Rural, Spring, Dry, Day, Car).

| Crashes | Lane Stable | Obstacle Avoidance | | | | |
|---------|-------------|-------------|----------|------|-------|-------|
| | | Pedestrian | Roadblock | Cone | Chair | House |
| No-FM | 7.3 /km | 8/10 | 9/10 | 10/10 | 9/10 | 10/10 |
| Ours | 0 /km | 0/10 | 0/10 | 0/10 | 0/10 | 1/10 |

Table 3: **Real Car Test**. We verify the generalization capability on a full-scale autonomous vehicle.

**Data Augmentation using Language.** In Table 2, we showcase the performance improvements achieved through data augmentation using language-augmented latent space simulation. Our procedure is as follows: (i) We first identify a set of target concepts likely to appear in the training data that are candidates for replacement, selecting *Tree* and *Dark* for this experiment; (ii) We then consult LLMs to suggest possible replacement concepts; in this experiments, they are broadly defined as *any non-drivable objects or entities likely to appear in a driving scenario*; (iii) Finally, we randomly swap image pixel features—those exhibiting high similarity to the target concepts—with the textual features corresponding to these suggested replacement concepts. We note performance improvements in most OOD scenarios, with the exceptions of *RSDNA* and *RSDNC*, where we observe a non-marginal decline in performance. Both represent rural, nighttime conditions, characterized by extremely low light, as depicted in row 3, column 2 of Tab. 1. These low-light environments make data augmentation particularly error-prone when targeting the concept of *Dark* for replacement.

**Real Car Deployment** In Tab. 3, we present the outcomes of tests conducted on a full-scale autonomous vehicle within a rural test track. These tests were carried out during the summer season and spanned various times of the day. Importantly, the evaluation took place on different road segments and occurred two years subsequent to the summer data used in the training set, allowing us to assess performance amid noticeable changes in the environment. We assess the system's proficiency in lane-following and its ability to avoid a variety of objects not encountered during training. These objects include pedestrians, roadblocks, traffic cones, chairs, and even a toy house. Some of these test scenarios are illustrated in the top row of Tab. 3. Our approach, which utilizes foundation models, yields near-flawless driving performance, further validating its effectiveness in generalizing to real-world robotic systems. However, it's worth noting that the inference speed is somewhat limited, averaging around 3 fps, compared to non-foundation-model-based policies, which achieve between 10 to 30 fps depending on the architecture. That said, we have not yet focused on optimizing runtime performance, which could be improved through techniques such as quantization.

## 4 Conclusion

This exploration into enhancing autonomous driving through multimodal foundation models has offered several lessons learned. The incorporation of these models improves the system's adaptability in unpredictable open-set environments, emphasizing their role in advancing real-world applicability of autonomous vehicles. Our model's blend of visual and textual understanding provides insights into the often murky decision-making processes inherent to autonomous systems, suggesting a promising trajectory for future models that prioritize both performance and transparency. The development of pixel/patch-aligned feature descriptors and latent space simulation, enriched with language modality, suggests potential for optimizing the training and debugging processes for end-to-end learning based control. Moreover, the successful deployment and performance of our methods on a real-world, full-scale autonomous vehicle provides encouraging first steps toward autonomous driving solutions that integrate multi-modal foundational models with perception.

**Acknowledgments**

This work is supported by Toyota Research Institute (TRI) and Capgemini Engineering. It, however, reflects solely the opinions and conclusions of its authors and not TRI or any other Toyota entity.

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

## A  Toward Generalization In End-to-end Autonomous Driving

The rapid technological advancement in autonomous driving has emerged as a pivotal innovation that shifts control from human hands to AI and sensors, promising safer roads, enhanced mobility, and unparalleled efficiency. In the pursuit of autonomous driving, end-to-end methodology offers a paradigm shift toward a holistic construction of the system that encompasses everything from perception to control. Such an approach has (i) more flexibility with minimal assumptions related to the design or functioning of sub-components, and (ii) better integrality toward an ultimate, unified goal in system performance evaluation and targeted optimization. Notably, the ongoing advancement in establishing end-to-end

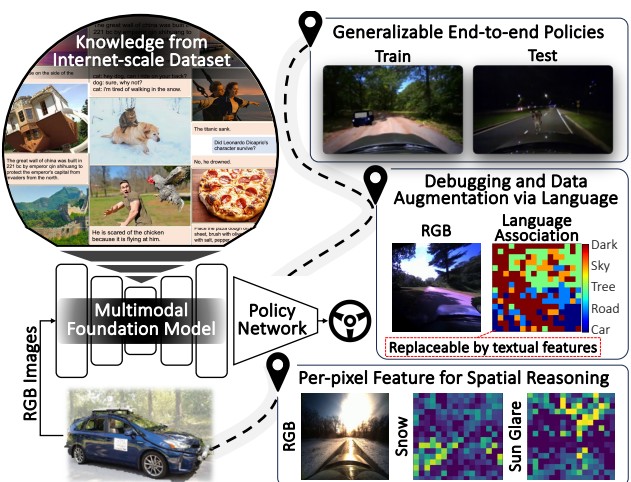

Figure 1: We harness the power of multimodal foundation models in end-to-end driving to enhance generalization and leverage language for data augmentation and debugging.

autonomous systems is propelled by the amalgamation of deep learning techniques by training models on extensively annotated datasets. However, prevalent systems exhibit the following prominent limitations.

**(i) Open set environments**: self-driving vehicles operate in extremely diverse scenarios that are impractical to fully capture within training datasets. When these systems encounter situations that deviate from what they've learned (i.e., out-of-distribution (OOD) data), performance deteriorates, giving rise to uncertainty and potential safety risks.

**(ii) Black-/gray-box models**: the ubiquitous use of complex, advanced machine learning models complicates the task of pinpointing the root causes of failures in autonomous systems. Unraveling the intricate interactions and identifying which learned concepts, objects, or even individual pixels contribute to incorrect behavior can be a daunting task.

On the positive side, deep learning is undergoing a transformative phase of significant advancement, characterized by the emergence of even larger and multimodal models [1, 5]. Trained on immense datasets that encompass billions of images, text segments, and audio clips, these models leverage knowledge gleaned from internet-scale resources to edge closer to achieving common-sense understanding. They have demonstrated exceptional efficacy in adapting to dynamic, open-set environments [16, 17]. In addition, the incorporation of language modality serves a dual purpose: not only does it offer an interface that is straightforwardly comprehensible by human users, but it also furnishes a concise yet rich representation of information that may sufficiently describe the underlying decision-making of autonomous systems.

## B  More Technical Details

**The end-to-end autonomous driving problem** involves designing a control system $\phi$ that produces steering and acceleration commands based on a continuous stream of perception data $F \in \mathbb{R}^{H \times W \times 3}$ (RGB imagery here), acquired through vehicle-mounted sensors $u = \phi(F)$. We propose enhancing $\phi$ by substituting the raw frames $F$ with a dense feature representation $F' \in \mathbb{R}^{H' \times W' \times D}$ extracted via a multimodal foundation model `Desc`, where $(H', W')$ is the resolution of the dense features in the spatial dimensions and $D$ is the number of channels, i.e., $u = \phi(F') = \phi(\text{Desc}(F))$.

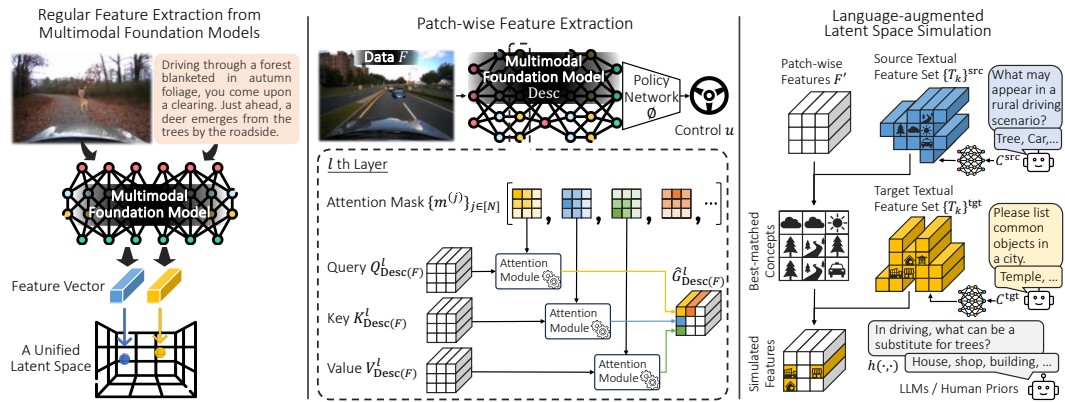

Figure 2: **Overview**. *Left:* Feature extraction from multimodal foundation models maps data in different modalities (e.g., image, text) to feature vectors in a unified latent space. *Middle:* We introduce a generic method for patch-wise feature extraction that preserves spatial information critical for end-to-end driving; this involves constructing attention masks anchored at each patch location to focus on specific regions (depicted by the coloring) for the attention module. *Right:* The multimodal representations with language modality enable seamless integration with LLMs; this allows to simulate latent features by substituting the original features $F'$ with contextually relevant language features (e.g., trees $\rightarrow$ house, shop, building).

**Setting the masks.** We define the $i$th entry of $m^{(j)}$ to correspond to "how much patch $i$ contributes to the semantic information of patch $j$". Analogous to convolutional kernels, "close" neighbors (patches) may contribute more than far ones. Let $(x_i, y_i)$ be the row-stacked ordering of the image grid after patching. Let $\text{dist}(i,j)$ denotes the distance between patch $i$ and $j$ as $\text{dist}(i,j) := \left\| (x_i, y_i) - (x_j, y_j) \right\|_z$, where $z \geq 1$ defines the norm. We set $m_i^{(j)} := f(\text{dist}(i,j))$; where $f$ can be $\begin{cases} 0, & \text{if } \text{dist}(0,1) > \alpha \\ 1, & \text{otherwise} \end{cases}$, $1/2^{\text{dist}(i,j)}$, or $1/\text{dist}(i,j)$, etc.

**Spatial resolution/number of patches $N$.** Increasing the spatial resolution enhances the foundation models' spatial features, as higher resolution allows for more granular, non-overlapping patches. For our applications, this granularity is beneficial. We adapt ViTs to extract overlapping patches during inference as in [13, 11], interpolating their positional encoding accordingly. This yields multimodal features with a finer spatial resolution notably without requiring additional training. In our empirical experiments, we have observed that this modification consistently performs well.

## C   Related Work

**End-to-end driving.** Neural networks trained to handle the entire process from perception to control in autonomous vehicles have displayed significant potential for maintaining various driving abilities [18, 19, 20, 9, 21]. Nevertheless, these networks encounter challenges in acquiring robust models on a large scale, as they demand extensive training data that is both time-consuming and costly to gather [8]. These challenges not only incur substantial expenses but also pose potential safety risks [22]. Consequently, training and assessment of robotic controllers in simulated environments have emerged as a viable alternative [8, 23, 24, 25]. However, even these simulated environments can not cover enough scenarios, making trained networks (systems) highly sensitive to scenarios that differ from their training data.

**Foundation models in robots.** Recent strides in robotics have embraced foundational models, showcasing their ability to interact adeptly in dynamic open-set scenarios, e.g., for control and planning [26, 27, 28, 29, 30], for 3D mapping [31, 32], detection and following systems [11, 33, 34, 35], and 3D scene segmentation and understanding [17, 16]. Moreover, these models have demonstrated their versatility across multiple data modalities [36, 37, 38, 39, 11, 16] marking a

new era of robots that can reason and interact wisely with the environment. Specifically in driving, explainable and language-based representations have been of interest for the ability to introspect and counterfactually reason about event [40, 41, 42, 43, 44]

**Pixel/patch aligned descriptors.** Several approaches for extracting per-pixel feature descriptors via foundation models were suggested [13, 16], however, they are either (i) not multimodal [13], (ii) trained with a specific focus on aligning foundation features with 2D pixels, and thus, these models tend to lose a substantial number of concepts as part of the fine-tuning process [45], or (iii) relies on the use of a universal segmentation model such as SAM [12], FastSAM [10], or Mask2Former [46] for extracting masks [16], and then applies the foundation models on crops of these masks to extract mask-aligned features, such methods are by definition inefficient for realtime applications and might yield not meaningful features when applying the foundation models on small crops. Finally, they might miss important regions in the image due to the used segmentor limitations.

## D  Experimental Setup

**Hardware setup.** We collected data and deployed learned policies on a full-scale vehicle (2019 Lexus RX 450H) retrofitted for autonomous driving. The car is equipped with an NVIDIA 4070 Ti GPU and an AMD Ryzen 7 3800X 8-Core Processor. For perception, we employ a 30Hz BFS-PGE-23S3C-CS camera offering a $130°$ horizontal FoV at a resolution of 960 x 600 pixels. Also, the car also features inertial measurement units (IMUs), wheel encoders, and an OxTS d-GPS system for precise odometry estimation.

**Tasks and evaluation metrics.** We focus on a generic driving task involving both lane-following and obstacle avoidance. Failure conditions are defined as: (i) veering off the lane boundary, (ii) colliding with objects (or approaching them too closely in the real world for safety reasons), and (iii) deviating from the lane's direction by more than $30°$. In simulations, we utilize a "soft" success rate as a performance metric, which gauges the duration the car can operate without encountering any failure conditions, normalized by a predefined time horizon for each trial. We conducted 100 trials, each with an approximate 20-second time horizon. For real-world testing, we tally the number of interventions made by the safety driver, using criteria that align with the aforementioned failure conditions. Unless stated otherwise, all experiments adhere to a closed-loop control setting.

**Data and learning.** We obtained data from a data-driven simulator VISTA [8] to augment a real-world dataset with diverse synthetic data in a closed-loop control setting. This simulation relies on approximately two hours of real-world driving data, gathered under varying conditions including different times of day, weather conditions, scenes, and seasonal variations. We employ a training approach known as Guided Policy Learning [47, 8], which takes advantage of privileged information within the simulator to guide the learning of the image-based policies. Ground-truth control signals for training are produced using a Proportional-Integral-Derivative (PID) controller for lane-following tasks and Control Barrier Functions (CBFs) [48] for obstacle avoidance. All models are trained with the Adam optimizer at a learning rate of $10^{-3}$, employing a plateau scheduler with a factor of 1 and a patience of 10, for total $10^6$ iterations. Our method uses BLIP2 [5] as it is SOTA in various benchmarks like zero-shot VQA, image-text retrieval, etc.

## E  More Results

**Cross-modality Generalization.** In Tab. 4, we evaluate the ability of our policy to generalize from image to language modalities. Notably, our policy is trained exclusively on image data. To generate cross-modality features, we employ the following procedure: (i) calculate features for a predefined set of natural language concepts (e.g., road, car); (ii) identify the best-matching textual feature for each image feature pixel, along with the degree of similarity; (iii) replace the image feature at pixels where the similarity exceeds a certain threshold with the corresponding textual feature. A null threshold implies driving solely based on textual features, while an infinity threshold denotes reliance exclusively on image features. Our observations indicate that the policy performs

| Thresh. | null | 0.03 | 0.05 | 0.08 | 0.10 | 0.20 | $\infty$ |
|---|---|---|---|---|---|---|---|
| **Perform.** | 0.657 | 0.658 | 0.661 | 0.751 | 0.826 | 0.984 | 1.000 |

Table 4: **Cross modality generalization.** We test generalization of training in image modality and deploying in language modality by replacing sufficiently similar image features (determined by the threshold) with textual features.

| Debugging Concepts | Perform. | Failure at | | |
|---|---|---|---|---|
| | | Lane Stable | Avoidance | Recovery |
| Image Feature | 1.000 | 0.000 | 0.000 | 0.000 |
| Car, Road, Tree, Sky | 0.267 | 0.149 | 0.453 | 0.117 |
| + Car & Road + Dark | 0.536 | 0.028 | 0.381 | 0.056 |
| + Car Exterior Parts | 0.657 | 0.040 | 0.237 | 0.074 |

Table 5: **A debugging tool**. We use LLMs to propose potentially relevant concepts for language-augmented latent space simulation to inspect the decision making of a policy.

reasonably well in cross-modality settings, both qualitatively and quantitatively. These results offer promising empirical evidence for the viability of language-augmented latent space simulations.

**Debugging and Inspecting Policies With Language.** In Tab. 5, we present a case study focused on policy debugging through language-augmented latent space simulation. Our procedure is as follows: (i) we consult Large Language Models (LLMs) to generate a base set of natural language concepts relevant to, for example, a rural driving scenario; (ii) we collect driving policy rollouts along with intermediate features, filtering out less pertinent concepts based on similarity statistics, and potentially human judgment; (iii) we then evaluate the policy by replacing image pixel features with textual features drawn from various subsets of these relevant concepts; (iv) lastly, we pinpoint specific concepts whose presence across subsets leads to significant performance changes. It's worth noting that the third step involves combinatorial complexity and may benefit from some degree of human intervention for enhanced efficiency. We highlight two major discoveries. Firstly, the concepts *Car & Road* and *Dark* play critical roles in both recovery and stable lane maneuver. Considering *Car* and *Road* in conjunction is essential, as the policy uses the end of the road as a navigational reference to maintain lane position. Typically, this reference point in the image is relatively small and encompasses both concepts simultaneously. Moreover, this combined feature serves as a natural differentiator, setting itself apart from standalone features like *Road* or *Tree* that may appear elsewhere in an image. As for the concept of *Dark*, it exposes a loophole where the policy takes advantage of simulation artifacts. Specifically, when the car deviates significantly from the lane center, the simulation produces dead or black pixels due to the absence of content to render.

