# OpenReview forum: "Drive Anywhere: Generalizable End-to-end Autonomous Driving with Multi-modal Foundation Models"
_robot-learning.org/CoRL/2023/Workshop/OOD — OOD Workshop @ CoRL 2023_

### Official Review · Reviewer_g1AL · 2023-10-14
**The paper proposes a method of feature extraction and language-based manipulation from multimodal foundation models.**

**Rating:** 7
**Confidence:** 3

**Review:**

Summary:

This work proposes two things:

(1) A method of extracting patch-aligned features from transformer-based vision models. This can be achieved for an arbitrary number of patches at arbitrary network layers by introducing an attention mask $m^{(j)}$, which determines how much information the desired feature patch $j$ should be take from other feature patches. The extracted features can then be used for downstream tasks, with experiments demonstrating competitive or improved performance on driving tasks.

(2) The use of the aforementioned feature extraction method can enable seamless integration with LLMs, where patch features can be replaced with alternative textual features to simulate different observations. This can be used for, e.g., data augmentation whereby concepts present in training data are swapped for possible alternatives (such as replacing the features that correspond to a tree with those that correspond to a house).

Autonomous driving experiments are performed in simulation and the real-world and outperform or demonstrate competitive performance with baselines.

Relevance:

This work is relevant to the workshop theme. The paper proposes a method of visual feature extraction/manipulation from foundation models, which the authors show can be used to improve a policy's performance in out-of-distribution (OOD) settings or augment existing training data to help generalize performance.

Writing Quality:

The paper demonstrates no outstanding grammatical or typographical errors. Clarity of presentation could be improved in certain areas (see below).

Novelty and Significance:

Visual foundation models (e.g., DINO, SAM, etc.) capture universal representations, however these the question of how to best use these representations for fine-grained visual understanding or policy learning.

Strengths:

The proposed method for feature extraction is relatively straightforward to implement and applicable to a variety of transformer-based visual foundation models. The method is quite flexible in that the patch-aligned features can represent arbitrarily-shaped regions at seemingly arbitrary resolutions. Experiments demonstrate that the features extracted in this can yield policies that are on par with, or outperform the comparison policies in OOD settings.

Additionally, feature replacement via the language modality is shown to be a possible approach to data augmentation through experiments which indicate improved generalization performance.

Weaknesses:

There are a few areas where additional detail would greaty improve comprehension and clarity:

-In Section 2, it is unclear how the attention mask $m^{(j)}$ is chosen. Is the mask manually specified or learned?

-Generally, it is unclear what the task entails in the experiment discussed in Section 3: Out-of-Distribution Generalization. It is understood to be an autonomous driving task, but a brief description would be helpful.

-What do the numerical figures indicate in Table 1?

-In Table 3, at first glance it appears that the paper's method fails on all obstacle avoidance tasks (everything appears 0/10), whereas the text indicates that performance was "near-flawless." A note in the caption or annotation in the table as to what the values refer to would greatly help in comprehension.

---

### Official Review · Reviewer_BePR · 2023-10-14
**Good ideas but writing needs to be improved**

**Rating:** 6
**Confidence:** 4

**Review:**

The paper proposes leveraging foundation models for extracting pixel-wise features, and incorporating latent simulation by replacing features with ones from LLM. Rich semantic features from foundation models allow better OOD generalization to complex weather and road conditions. Experiment results clearly demonstrate the benefits of the approach in improving OOD generalization performance. The supplementary video also conveys the high-level ideas well and shows sample experiment settings.

While the idea of harnessing foundation model semantics is simple and intuitive, the paper at the current stage does not convey the full approach well. I understand that the paper seems to have moved a lot of details in the appendix, but the current main text does not explain well (1) the overall policy architecture, (2) dataset used, (3) loss function used. Thus the approach and experiment sections are difficult to read and out of context. These details should better be kept in the main text. Even with the appendix (Appendix D, Experiment Setup), some details such as the exact training procedure are still unclear. For example, how is Guided Policy Learning applied in the work?

Also, I find the approach section about extracting features difficult to follow: it would be better to include a partial version of Figure 1 and Figure 2 in the main text.

---

### Decision · Program_Chairs · 2023-10-17

**Decision:**

Accept

**Comment:**

We agree with the reviewers’ assessment that this work is technically sound and will contribute to productive, topical discussions at the 2023 Workshop on OOD Generalization in Robotics. In particular, we appreciate the clear improvement demonstrated in OOD generalization performance, and agree with the reviewers that the definition of OOD in this context (i.e., task description) should be made as clear as possible. We recommend the authors incorporate the reviewers’ feedback into their camera-ready submission to further improve their manuscript.